# Genome-Wide Identification of Auxin Response Factors in Peanut (*Arachis hypogaea* L.) and Functional Analysis in Root Morphology

**DOI:** 10.3390/ijms23105309

**Published:** 2022-05-10

**Authors:** Lu Luo, Qian Wan, Zipeng Yu, Kun Zhang, Xiurong Zhang, Suqing Zhu, Yongshan Wan, Zhaojun Ding, Fengzhen Liu

**Affiliations:** 1Key Laboratory of Plant Development and Environmental Adaptation Biology, Ministry of Education, School of Life Sciences, Shandong University, Qingdao 266237, China; luluo0920@163.com (L.L.); yzp52120090916@163.com (Z.Y.); 2State Key Laboratory of Crop Biology, Shandong Key Laboratory of Crop Biology, College of Agronomy, Shandong Agricultural University, Tai’an 271018, China; kunzh@sdau.edu.cn (K.Z.); zhangxr_1987@sina.com (X.Z.); zhusuqing0708@163.com (S.Z.); yswan@sdau.edu.cn (Y.W.); 3State Key Laboratory of Crop Biology, Shandong Key Laboratory of Crop Biology, College of Life Science, Shandong Agricultural University, Tai’an 271018, China; wanqian9336@163.com

**Keywords:** peanut, auxin response factor, genome-wide identification, expression, root morphology

## Abstract

Auxin response factors (ARFs) play important roles in plant growth and development; however, research in peanut (*Arachis hypogaea* L.) is still lacking. Here, 63, 30, and 30 *AhARF* genes were identified from an allotetraploid peanut cultivar and two diploid ancestors (*A*. *duranensis* and *A*. *ipaensis*). Phylogenetic tree and gene structure analysis showed that most *AhARF*s were highly similar to those in the ancestors. By scanning the whole-genome for ARF-recognized *cis*-elements, we obtained a potential target gene pool of AhARFs, and the further cluster analysis and comparative analysis showed that numerous members were closely related to root development. Furthermore, we comprehensively analyzed the relationship between the root morphology and the expression levels of *AhARF*s in 11 peanut varieties. The results showed that the expression levels of *AhARF14*/*26*/*45* were positively correlated with root length, root surface area, and root tip number, suggesting an important regulatory role of these genes in root architecture and potential application values in peanut breeding.

## 1. Introduction

Auxin is a key hormone that plays an important role throughout the life cycle of plants from embryogenesis to fruit maturity [1,2,3]. Auxin response factors (ARFs) play a core role in auxin-mediated transcriptional regulation and auxin signaling transduction 2a ARF proteins contain three conserved domains. Briefly, the *N*-terminal DNA-binding domain (DBD) is responsible for recognizing the auxin-recognized cis-element (AuxRE) within target gene promoters [4,5]. The *C*-terminal Phox and Bem1 (PB1) domain mediates ARF–ARF or ARF–Aux/IAA (Auxin/INDOLE ACETIC ACID) dimerization. The middle domain determines whether ARF is a transcription activator or a transcription suppressor due to the variety of amino acids it contains [6,7]. At low-auxin levels, Aux/IAAs inhibit ARF transcriptional activity through recruit TOPLESS (TPL) and histone deacetylase (HDAC) [8]. With the increase of auxin and the degradation of Aux/IAA through the SCFTIR1-dependent pathway, the released ARF transcription factors regulate the expression of downstream targets and cause auxin-related plant growth and development [9,10].

There are multiple ARF members in different plants. For example, 23 in *Arabidopsis*, 25 in rice, and 31 members in maize [11,12,13]. Importantly, the role of some ARF-mediated auxin signaling in root growth and development has been well studied in both the model plant *Arabidopsis* and some crops. For example, ARF10/16 has been shown to regulate root cap cell formation and root stem cell identity in *Arabidopsis* [14]. ARF7/19-mediated auxin signaling promotes lateral root development by inducing the expression of LATERAL ORGAN BOUNDARIES DOMAIN (LBD16/18/29) and/or PR-1 homolog (PRH1) in *Arabidopsis* [15,16,17]. In soybean, GmARF8a/b negatively regulates lateral root development and nodule formation [18]. SlARF2 promotes lateral root formation in tomato [19]. Although ARFs appear to be conserved in seed plants, the question remains whether the functions of these proteins are also consistent across plant species. Therefore, it is necessary to analyze the expression pattern, DNA-binding specificity, and the relationship with important traits, such as root morphology, before replications in breeding [20,21]. 

Peanut (*Arachis hypogaea* L.) is one of the most important economic oil crops, grown in many Asian and African countries with an annual production of about 46 million tons (http://www.fao.org/faostat; accessed on 2 May 2021). Cultivar peanut (*A. hypogaea*, allotetraploid, AABB, 2n = 4x = 40) is assumed from hybridization of two wild diploid ancestors, *Arachis duranensis* (AA genome, 2n = 2x = 20) and *Arachis ipaensis* (BB genome, 2n =2x =20) [22]. Peanut cultivars exhibit extensive phenotypic and genetic variation, especially the root traits [23,24]. However, due to limited research methods, the regulation genes of important root traits are rarely identified in peanut. Their application in breeding is seriously lagging compared with other crops. Despite the importance of ARF proteins in root development, the role of peanut ARFs has not been characterized. Here, we identified 63 *AhARF*, 30 *AdARF*, and 30 *AiARF* genes from an allotetraploid peanut cultivar, *A. duranensis*, and *A. ipaensis*, respectively. Comprehensive analysis revealed that numerous predicted downstream targets of AhARFs are enriched in root development, especially *AhARF14*/*26*/*45*. Importantly, the expression of these genes is highly correlated with root architecture, implying the involvement of these members in root growth and development. 

## 2. Results

### 2.1. Genome-Wide Identification of Peanut ARF Genes

Based on a HMMSCAN and BLASTP search, 30, 30, and 63 *ARF* genes were identified in *A*. *duranensis* (*AdARF*s), *A*. *ipaensis* (*AiARF*s), and *A. hypogaea* (allotetraploid peanut cultivar; *AhARF*s), respectively. These *AhARF* genes were numbered according to their locations in different chromosomes, and the *AdARF*s and *AiARF*s were named according to their gene IDs (Figure 1). Detailed information for all these genes, including gene ID, chromosome location, molecular weight (MW), and the isoelectric point (pI) of proteins are listed in Appendix A. 

There were 62 *AhARF* genes mapped to 18 chromosomes (Figure 1), and *AhARF63* is located on scaffold 59. Similar to those of *A. hypogaea*, there was no *ARF* gene mapped to chromosome Aradu.A01 or Araip.B01. The number of *AhARF*s on each chromosome was approximately the same as that of the corresponding chromosome of *A. duranensis* and *A. ipaensis*. The absence of *ARF*s located on chromosomes Arahy.01 and Arahy.11 was also consistent with that of *A. duranensis* and *A. ipaensis*. 

### 2.2. Phylogenetic Analysis of Peanut ARFs

An unrooted maximum-likelihood phylogenetic tree was constructed using ARFs from an allotetraploid peanut and its diploid ancestors. In most cases, a pair of ARFs from each ancestor and a pair of AhARFs clustered to a clade, and, usually, the two ancestor ARFs, showed a closer evolutionary relationship (green and blue dots in Figure 2). In addition, the gene structure of most *AhARF*s and their orthologs has a high similarity (Figure 2), mainly reflected in the number and position of introns. Interestingly, according to the annotation information of *A. hypogaea* cv. Tifrunner genome, alternative splicing (AS) events were identified in 15 peanut *ARF*s (Appendix A). A few AS events occurred in the untranslated regions (UTRs), which did not affect the protein structure. In fact, most AS events altered the coding of proteins and even led to a premature termination of translation (Appendix A). For example, the seventh and eighth exon of the AS1 of *AhARF1* (arahy.Tifrunner.gnm1.ann1.TZ3SVZ.1) were not identified in other AS events, while the ninth exon corresponded to two exons in other AS events. There were five and four possible AS events identified in *AhARF27* and *AhARF60*, respectively, and four of them (two for each) coding proteins with large *C*-terminal deletions. Considering that the *C*-terminal domain of ARFs is responsible for dimerization with other ARFs or Aux/IAA, these truncated ARFs may play different roles in planta. For example, AtARF3, an atypical ARF lacking the PB1 domain, can directly induce downstream gene expression in the presence of auxin, a more concise auxin transcriptional regulation mode.

To clarify which developmental or hormonal signals these *ARF*s are involved in, we then examined which *cis*-acting elements were present in their promoters. According to the prediction of a database search of plant promoter results, obtained from the PlantCARE website, auxin response elements were detected in the promoters of 42 peanut ARF genes, including 23 *AhARF*s, 9 *AdARF*s, and 10 *AiARF*s (Figure 2, black stars). In addition, other *cis*-elements were also found to participate in plant hormone responses and stress responses (Figure 2, inner cycle). In addition, species and the positions of *cis*-elements in genes belonging to the same clade are similar. *AdARF*s and *AiARF*s are similar to *AhARF*s, implying that this family is evolutionarily conservative. 

The function of AhARFs can also be predicted based on the well-characterized function of their homologous proteins in other species, such as *Arabidopsis thaliana*, *Oryza sativa*, *Solanum lycopersicum*, and *Glycine max* [4,6,14,15,16,17,18,19,20,21,22,24,25,26,27,28,29,30,31,32,33,34,35,36,37,38,39,40,41,42,43,44,45,46,47,48,49,50,51,52,53,54,55,56,57,58,59,60,61,62,63,64,65,66]. The phylogenetic analysis and function prediction results showed that most AhARFs were clustered with plant ARFs (Appendix A). These results revealed that the functions of ARFs in different species might be conserved, and the well-functioning-identified ARFs in other species have an important reference value in peanut. 

### 2.3. Gene Duplication on the Expansion of ARF Genes

The important regulatory roles of gene duplication in evolution and the expansion of gene families and crop domestication have been well documented. To clarify the evolution process of the *AhARF* family after the formation of the allotetraploid peanut, we analyzed the gene duplication events of *AhARF*s. As shown in Figure 3A and Appendix A, there were 22 orthologous, 14 paralogous, and 5 tandem gene pairs detected in *A. hypogaea*. 

The substitution rates of nonsynonymous (Ka) versus synonymous (Ks) were subsequently calculated to explore the selection pressure of *AhARF*s, and the results showed that the Ka/Ks values of all orthologue pairs were less than 1.0, and 88% (22/25) were less than 0.5 (Table 1). This suggests that the orthologous *AhARF*s undergo intensive purifying selection pressure and remain conserved in both structure and function. The divergence time of these five gene pairs (*AhARF40***–***AhARF61*, *AhARF28***–***AhARF40*, *AhARF37***–***AhARF56*, *AhARF36***–***AhARF57*, *AhARF39***–***AhARF58*) were estimated to be around 85–107 million years ago (Mya), while the other pairs were predicted to have diverged nearly 1.32–7.93 Mya. 

### 2.4. Expression Profiles of AhARFs 

To explore the possible biological functions of *AhARF*s, their expression patterns were analyzed based on RNA-seq data obtained from PeanutBase (https://www.peanutbase.org/ accessed on 21 April 2021). The *AhARF*s exhibited diverse expression patterns in different developmental stages and plant tissues; however, most of the gene pairs showed a similar expression pattern and were also clustered together in the heatmap (Figure 4). For example, *AhARF16* and *AhARF48* are highly similar in sequence and gene structure (Figure 1), and similarly, their expression patterns are almost identical in 22 plant tissues and at different developmental stages. However, there are exceptions, such as *AhARF1* and *AhARF31*, which are highly homologous but are expressed differently in peg and fruit development. 

### 2.5. Characterization of Downstream Target Genes of AhARFs

To predict the possible roles of these ARFs, we examined the distribution and function of target genes that may be regulated by these transcription factors. DR5 and IR8, two composite motifs, have been shown to be closely involved in the auxin-mediated transcriptional regulation. Therefore, we searched DR5, IR8, and single AuxRE (TGTCGG) motif in the 2000 bp upstream of each gene throughout the peanut genome (Figure 5A). There were 24,380 possible ARF-binding elements detected in 18,190 genes; among them, single AuxRE (TGTCGG) motif accounts for 65.44% (Appendix A). There were also 8426 DR5 or IR8 in 7350 gene promoters, and most of these genes contain only one DR5 or IR8 (Appendix A). The highest number of ARF-binding elements (DR5 and IR8 only) was found in the promoter of a predicted glucose-induced degradation protein (arahy.7YHY0R). 

Subsequent gene ontology (GO) analysis showed that the predicted target genes mainly participate in plant hormone responses, biotic/abiotic stress responses, and tissue/organ development (Figure 5C). Besides being involved in auxin polar transport and auxin response, these targets also relate to seven other well-characterized plant hormones, including abscisic acid (ABA), salicylic acid (SA), brassinosteroids (BRs), jasmonic acid (JA), ethylene (ETH), gibberellins (GAs), and cytokinins (CKs). Notably, the function of these genes is significantly enriched in plant root development, especially root hair elongation and root meristem growth (Figure 5C).

### 2.6. The Possible Regulatory Roles of AhARFs in Peanut Root Morphology 

To explore the relationship between the expression of *AhARF*s and the root morphology of peanut, we selected 11 peanut varieties (Appendix A) and analyzed their root architecture and the expression levels of *ARF*s in their roots. Scanning the root systems of different varieties revealed variable root architectures among them (Figure 6A). For example, JuHua7 (JH7), EHua3 (EH3), and ShanHua11 (SH11) display a long primary root and less lateral roots, while FengHua 2 (FH2) and Jinkins Jumbo (JK) exhibit a long primary root with more lateral roots. After quantifying the total root length, root surface area, root mean diameter, and root tip number, we found that these indexes also differed among varieties (Figure 6B). 

According to the expression patterns of *AhARF*s (Figure 4), seven members (*AhARF8*/*14*/*26*/*38*/*39*/*45*/*56*) were selected and their expression levels were further examined in the roots of these 11 varieties, including JK, Meiyinxuan 41165 (41165), Shixuan 64 (SX64), Juhua 27 (JH27), FH2, EH3, SH11, and Zhonghua 12 (ZH12). As shown in Figure 6C, these *AhARF*s exhibited different expression levels in different varieties, suggesting a possible link between the gene expression levels and root architecture. Among them, the expression of *AhARF59* showed the greatest difference among different varieties. Subsequently, the Pearson’s correlation analysis was performed to explore the relationship between root morphology and the expression levels of the *AhARF*s (Figure 6D). The results showed that the expression levels of *AhARF26* and *AhARF45* are positively correlated with total root length; *AhARF14*/*26*/*45* are positively correlated with root surface area; and *AhARF26* is positively correlated with the root tip formation, indicating the important role of these *AhARF*s in peanut root development.

Subcellular localization of AhARF14 and AhARF26 proteins were further analyzed in *Nicotiana benthamiana* (*N. benthamiana*) epidermal cells. AhARF26 was specially localized in the nucleus, while AhARF14 mainly localized in the nucleus and weak signal of AhARF14-GFP, which was also detected in the cytoplasm (Figure 7). The subcellular localization of AhARF14 was further analyzed using the online tool DeepLoc-1.0, and it showed a likelihood of 0.587 to localize in cytoplasm (Appendix A).

## 3. Discussion

A previous study reported that there were 114, 28, and 28 *ARF*s in *Shitouqi* (allotetraploid peanut cultivar), *A. duranensis*, and *A. ipaensis*, respectively [67]. Another independent study identified 61 *AhARF*s from *A. hypogaea*, and then revealed the regulatory roles of *AhARF6* in pod development [68]. Here, we supplemented and improved various data on this family, identified 63 members of *A. hypogaea*, and picked up the two missing members of *AhARF1* and *AhARF63*. Multiple sequence alignment showed that the three *AhARF1* AS formats and *AhARF63* belong to the typical *ARF*s. In addition, we have provided some valuable information on the upstream regulation of *cis*-elements and downstream targets. 

More importantly, we analyzed the potential relationship between the expression levels of *AhARF*s and the root morphology among varieties and found that expression levels of *AhARF14*/*26*/*45* are positively correlated with peanut total root length, root surface area, or root tip number. Peanut is an important protein source and oil crop that can be grown in poor quality soil [69,70], while peanut yield is greatly restricted by adverse environmental conditions, such as drought, salt, and nutrient deficiencies [71,72,73]. Roots are the main organs of plants to absorb water and nutrients from soil. Therefore, root morphology and their spatial configuration will substantially determine the ability of a plant to secure edaphic resources [74]. Understanding the molecular mechanisms that regulate crop root morphology is conducive to improving root traits and increasing crop yield. Root morphology is a complex trait determined by both environmental and endogenous factors, and among them, auxin plays an important role [75,76,77,78]. Therefore, genome-wide identification and function prediction are essential to unlock the mechanisms of *AhARF*-mediated auxin signaling in peanut root development. 

Root morphology traits have long been considered a key target by breeders for crop improvement; however, there are many challenges in root system measurement [79]. Although several root trait QTLs have been observed via analysis of crop populations, important genes are rarely reported [80,81,82,83,84]. Most favorable root traits are selected directly by visual test in breeding practices. The selection of germplasms represents better root traits, and finding their commonalities in genotypes or expression patterns is a promising practice. Through RNA-seq analysis, we found that seven *AhARF*s were highly expressed in roots, implying the involvement of these members in root growth regulation. Importantly, we showed that the expression levels of *AhARF14*/*26*/*45* are largely positively correlated with root architecture. These three peanut genes exhibit high similarity with their homologous *AtARF16*, which has been reported to control root cap formation [14]. *AhARF8*/*38*, which were not related to the peanut root traits tested in our work, seem to display different functions from their homologous *AtARF7*/*19*. These results suggest that, although ARFs are conserved in different plant species, their functions are diversified. The identification of germplasms with effective genotypes and linked molecular markers that regulate root development is beneficial to molecular marker-assisted breeding. The finding that *AhARF14*/*26*/*45* modulate peanut root architecture links their expressions to beneficial root traits, and it can be used directly in the selection of parental germplasms and their hybrid offspring, especially for drought-resistance and nutrition-efficient breeding. Interestingly, a weak AhARF14-GFP signal was detected in cytoplasm, which is inconsistent with previous studies on the subcellular localization of plant ARF proteins [68,85]. This may be caused by differences between species and may indicate new regulatory mechanisms for ARFs in peanut.

## 4. Materials and Methods

### 4.1. Plant Materials and Growth Conditions 

The *A. hypogea* L. cultivars and germplasms used in this study were selected from a natural peanut population with extensive phenotypic and genetic variation preserved by our group [23]. Mature seeds were soaked in 0.1% H_2_O_2_ for 6 h, and then washed 3–5 times with sterile ddH_2_O. Sterilized seeds were sown on water-wet degreasing cotton, extended in seedling cultivation disks to germinate at 26 °C in darkness for 3 days. Seedlings were then exposed to long-day conditions (16 h light and 8 h dark cycle, 11.4 K LuX light intensity) for another 2 or 3 days. Two-functional-leaf seedlings were transplanted to the hydroponic box and cultured with 1/5 Hoagland’s nutrient solution until further use [86].

### 4.2. Genome-Wide Identification of AhARFs, AdARFs, and AiARFs

Whole-genome sequences of allotetraploid peanut *A. hypogaea* cv. Tifrunner (version 1) [87] and the diploid ancestors (*A. duranensis* and *A. ipaensis*) [79] were downloaded from PeanutBase (https://www.peanutbase.org/peanut_genome; accessed on 21 April 2021 ). The protein sequences of AtARFs were downloaded from the TAIR database (http://www.arabidopsis.org/, accessed on 21 April 2021), while those of rice, soybean, *Medicago truncatula*, and tomato were downloaded from Ensembl Plants (http://plants.ensembl.org/index.html; accessed on 21 April 2021). The hidden Markov model (HMM) profile of the B3-type DNA-binding domain (DBD, PF02362), auxin-response domain (PF06507), and Aux-IAA domain (PF02309) were retrieved from the Pfam database (http://pfam.xfam.org/; accessed on 15 March 2021) and used for peptide-searching through BLASTP analysis. The predicted peanut ARFs were confirmed using the SMART program (http://smart.embl-heidelberg.de/; accessed on 21 April 2021) and the Conserved Domain Database (CDD; https://www.ncbi.nlm.nih.gov/cdd; accessed on 11 April 2021). The protein sequences of peanut ARFs were submitted to the ProParam tool in ExPASy (https://web.expasy.org/protparam/; accessed on 30 April 2021) to calculate the molecular weight (MW) and theoretical isoelectric point (pI).

### 4.3. Chromosome Distribution, Gene Structure, Phylogenetic Analysis, and cis-Elements Analysis 

Information on peanut *ARF*s loci on chromosome were derived from annotation gff3 files downloaded from PeanutBase (https://www.peanutbase.org/; accessed on 18 April 2021). The Gene Structure Display Server (https://gsds.cbi.pku.edu.cn/; accessed on 18 April 2021) was employed for analyzing *AhARF* structure according to the gff3 data. MEGA7 [88] was used to perform sequence alignment and maximum-likelihood phylogenetic tree construction with the bootstrap method (number of bootstrap replications = 1000). To analyze the regulatory region of the *AhARF*s, the 2000 bp genomic sequences located upstream of the start codon ATG were analyzed in the PlantCARE website (http://bioinformatics.psb.ugent.be/webtools/plantcare/html/; accessed on 18 April 2021). Visualization and further editing of the phylogenetic tree were performed on the website tool iTOL (http://itol.embl.de/; accessed on 25 April 2021).

### 4.4. Syntenic and Genome Duplication Analysis of Peanut ARFs

The Multiple Collinearity Scan toolkit (MCScanX) was used to identify gene duplication events according to a previous study [89]. The amino acid sequences of duplicated *ARF* pairs were aligned first, and then used to guide the alignment of cDNA sequences with in-house Perl-scripts. The nonsynonymous (Ka) and synonymous (Ks) substitution ratios were calculated by the Ka/Ks calculator using the YN model. The divergence times (Mya) of *A. hypogaea* and the diploid ancestors (*A. duranensis* and *A. ipaensis*) were calculated with the formula T = Ks/2r. The r (neutral substitution rate) was taken to be 8.12 ×10^–9^ according to a previous study [79].

### 4.5. Genome-Wide Search for Downstream Targets of AhARFs

Upstream regions (2000 bp upstream of the start codon) of genes in the peanut genome were used for screening. Searching of the AuxRE (DR5 and IR8) was performed on MEME Suite (Motif-based sequence analysis tools, http://meme-suite.org/; accessed on 13 May 2021) using Motif Cluster Alignment and Search Tool (MCAST) [90,91]. The match *p*-value was set at <10^–4^. The peptide sequences were downloaded from PeanutBase. The GO annotation of genes carrying AuxRE repeats was performed using OmicsBox (Version 1.4.11). 

### 4.6. Expression Analysis of AhARFs

The expression atlas of 22 *A. hypogaea* tissues was downloaded from PeanutBase (https://www.peanutbase.org/gene_expression/atlas; accessed on 21 April 2021) [79]. In these RNA-Seq data, the normalized reads were mapped to an in silico amphidiploid genome assembled from the genome of the diploid ancestors *A. duranensis* and *A. ipaensis* [92]. BLAST was performed to identify the homologous genes of *AhARF*s in *A. duranensis* and *A. ipaensis*. Only the *AdARF*s or *AiARF*s showed the highest similarity in amino acid sequence with *AhARF* was defined as homologous gene of *AhARF*. The IDs of the homologous gene were used to extract the fragments per kilobase of transcript per million mapped reads (FPKM) values from the tissue expression atlas.

Quick RNA isolation kit (Waryong, Beijing, China) was used to isolate the total RNA following the manufacturer’s instructions. Samples were quantified by NanoDrop 2000 microvolume spectrophotometry (Thermo, Wilmington, DE, USA), and 1 μg RNA was reverse-transcribed by the PrimeScript RT reagent Kit with DNA Eraser (Takara Bio, Dalian, China) following the manufacturer protocol. Total RNA and cDNA were stored at −80 °C and −20 °C, respectively. qRT-PCR was performed on an ABI StepOne Real-Time PCR Systerms (Thermo, Wilmington, DE, USA) using SYBR Premix Ex Taq (Takara, Dalian, China). Details of primers are provided in Appendix A. *AhUKN1* was used as the internal reference [93]. Three technical replicates were included in each biological replicate, and three biological replicates were performed. 

### 4.7. Root Morphology Analysis

For root morphology analysis, roots of 21-day-old seedlings were harvest and washed with deionized water. The WinRHIZO LA-S image analysis system (WinRHizo LA-S, Regent Instr. Inc., Quebec, Canada) was used for scanning and quantization of total root length (TRL), root surface area (RSA), root mean diameter per plant (ARD), and root tip number (RTN). Three independent biological replications were performed for each germplasm, and three technical repeats were included in each biological replication. For each technical repeat, roots from 6 plants were scanned and analyzed.

### 4.8. Statistical Analysis

GraphPad Prism version 8.0.2 (GraphPad, San Diego, CA, USA) was used to prepare the figures. Data analysis was performed by SPSS 21.0 (SPSS Inc., Chicago, IL, USA) using one-way ANOVA. Differences were considered significant at a probability level of *p* < 0.05.

### 4.9. Subcellular Localization Analysis

The full-length CDS of *AhARF14* and *AhARF26* were amplified from JK using cDNA as a template. Primers used are listed in Appendix A. The purified PCR products were cut by *BamH* I and *Sal* I (TaKaRa, Dalian, China) and cloned into pBI122 vector. The sequenced pBI122-35S::AhARFs-GFP vectors were transformed into *Agrobacterium tumefaciens* strain GV3101 and coinfected into *N. benthamiana* leaf epidermal cells. Plants were grown at 26 °C for 48–60 h, and GFP signals in the leaves were detected at 488 nm using an LSM 51 confocal laser scanning microscope (Carl Zeiss, Jena, Germany). The pBI122-35S::GFP vector was used as a control. Three independent biological repeats were performed. Subcellular localization of AhARF14 and AhARF26 was predicted using online tool DeepLoc-1.0 (https://services.healthtech.dtu.dk/service.php?DeepLoc-1.0; accessed on 21 January 2022). 

## Figures and Tables

**Figure 1 ijms-23-05309-f001:**
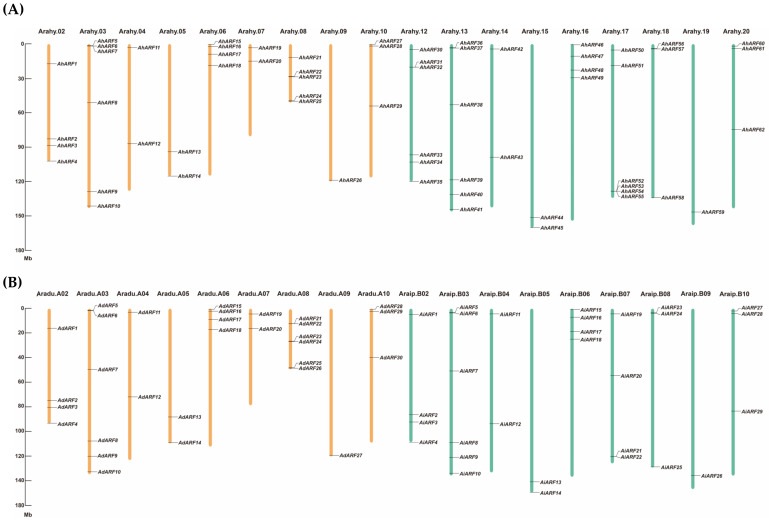
Chromosome distribution of ARFs in cultivar (**A**) and wild (**B**) peanuts. Chromosomes belong to the A sub-genome of *A. hypogaea* (Arahy.02–10) and their homologues from *A. duranensis* are colored in yellow, while the chromosomes that belong to the B sub-genome of *A. hypogaea* (Arahy.12–20) and their homologues from *A. ipaensis* are colored in green. Chromosomes that do not contain *ARF*s and *ARF*s located on the scaffold are not shown.

**Figure 2 ijms-23-05309-f002:**
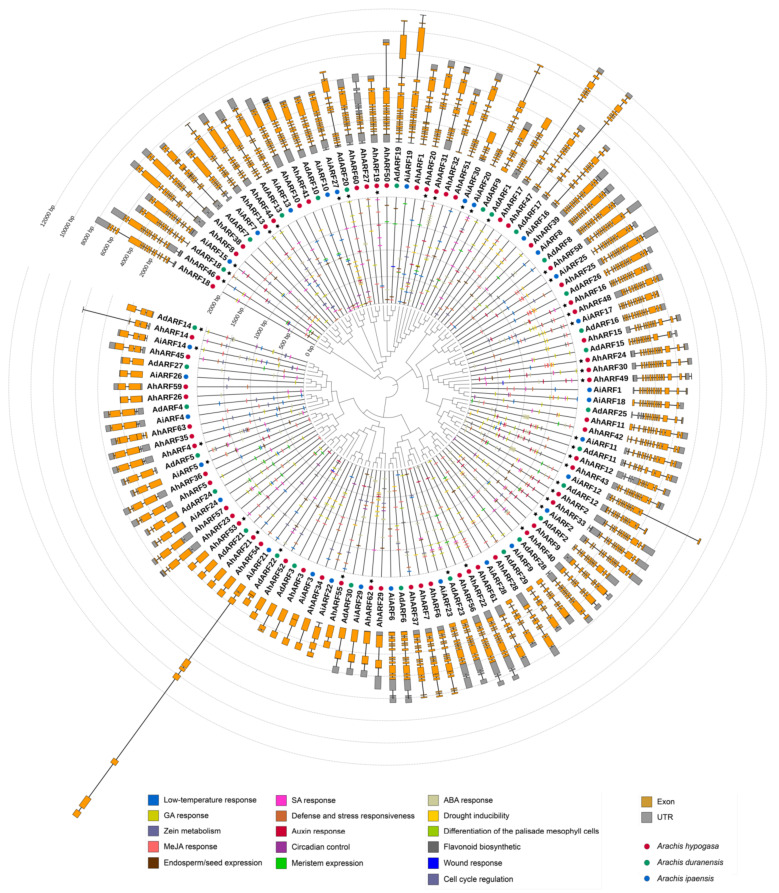
Phylogenetic tree, *cis*-elements, and gene structure analysis of *ARF*s in wild and cultivar peanuts. Gene structures are shown in the outer cycle, and *cis*-elements are shown in the cycle inside the gene names. The red, green, and blue dots represent *AhARF*s, *AdARF*s, and *AiARF*s, respectively.

**Figure 3 ijms-23-05309-f003:**
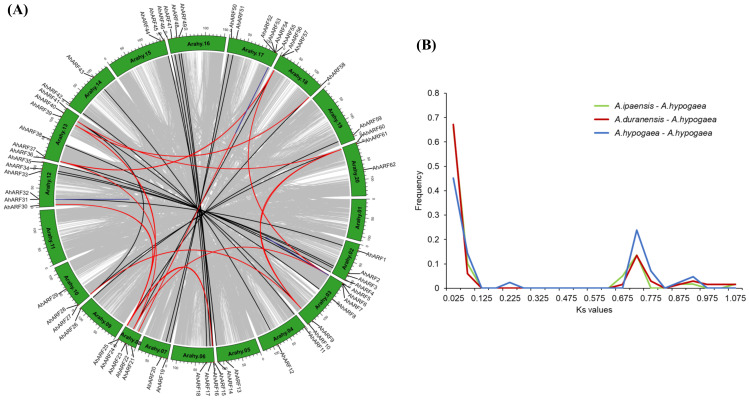
Duplication derived *ARF*s in wild and cultivar peanuts. (**A**) The orthologous genes (black lines), paralogous genes (red), and tandem repeat (blue) in cultivar peanut. (**B**) The Ks value distribution of the duplicated orthologous genes between the wild and cultivar peanut.

**Figure 4 ijms-23-05309-f004:**
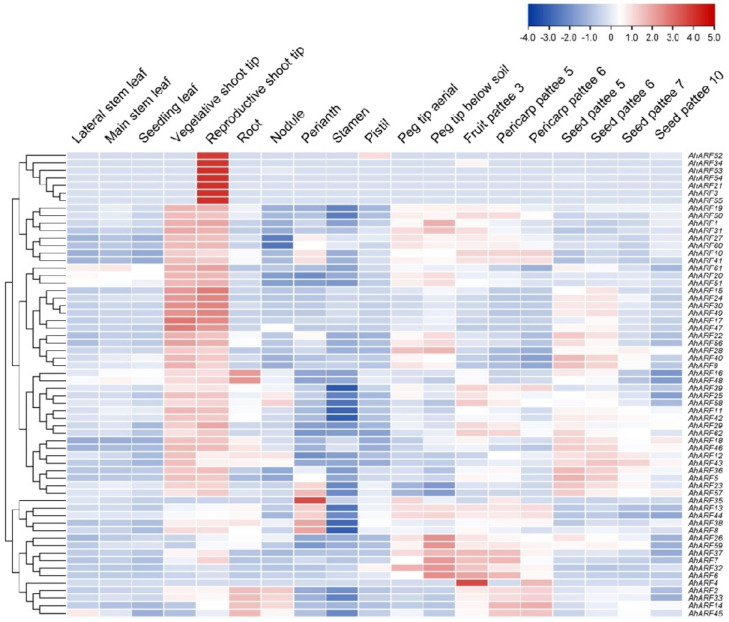
Heatmap illustration of tissue expression patterns of *AhARF*s based on FPKM values. The log2 transformation of FPKM values and visualization were performed by TBtools. Scalebar on top right indicates the levels of gene expression.

**Figure 5 ijms-23-05309-f005:**
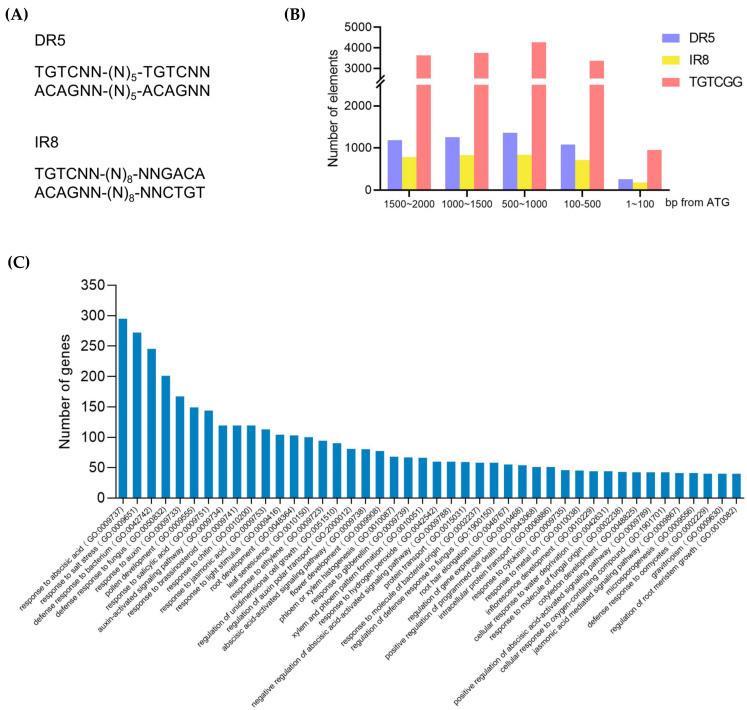
Genome-wide prediction of ARF-binding elements. (**A**) Definition of DR5 and IR8 elements. N indicates A, C, G, or T. (**B**) Number of DR5, IR8, and AuxRE in peanut genomes. (**C**) GO analysis of genes carrying DR5, IR8, or AuxRE in their promoters, and only the GO terms of those containing at least 40 genes are shown here.

**Figure 6 ijms-23-05309-f006:**
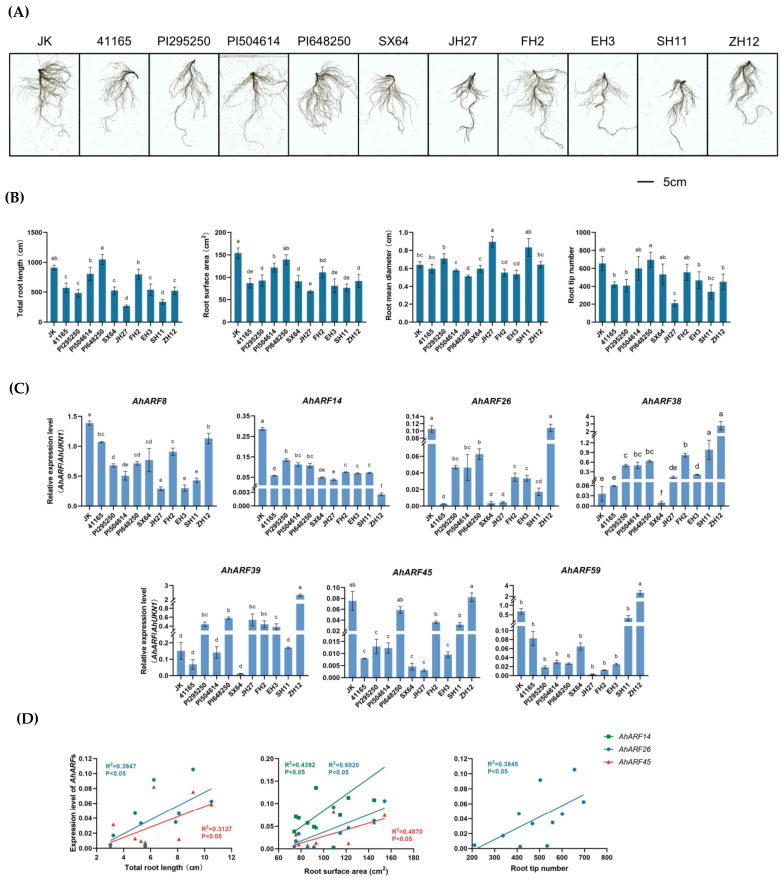
Root morphology and their correlation with *AhARF* expression levels in peanut germplasms. (**A**) Root morphology of peanut germplasms. JK, Jinkins Jumbo; 41165, Meiyinxuan 41165; SX64, Shixuan 64; JH27, Juhua 27; FH2, Fenghua 2; EH3, Ehua 3; SH11, Shanhua 11; ZH12, Zhonghua 12. (**B**) Total root length, root surface area, root mean diameter, and root tip number of peanut germplasms. Lower case letters (a, b, c, d) indicate statistically significant differences between cultivars. (**C**) Relative expression levels of *AhARF*s in peanut roots. (**D**) Pearson’s correlation analysis between root morphology and expression of *AhARF*s. R^2^, Pearson’s correlation coefficient.

**Figure 7 ijms-23-05309-f007:**
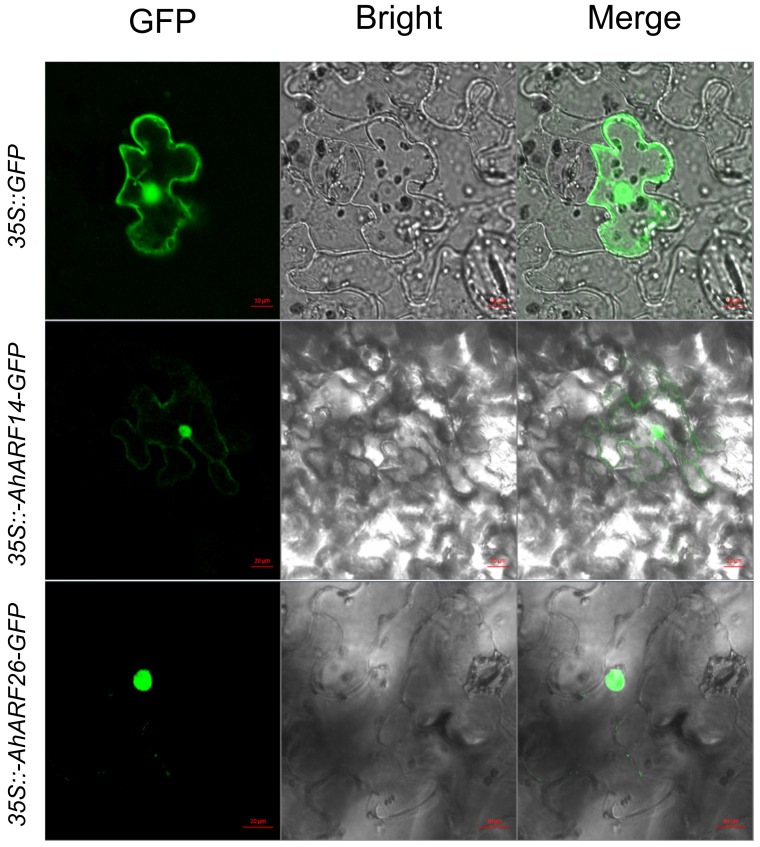
Subcellular localization analysis of AhARFs in *N. benthamiana* leaves.

**Table 1 ijms-23-05309-t001:** The nonsynonymous (Ka) and synonymous (Ks) substitution rates of orthologue AhARF pairs in *A. hypogaea*.

Gene 1	Gene 2	Ka	Ks	Ka/Ks	Divergence Time (Mya)
*AhARF3*	*AhARF34*	0.0377	0.0517	0.7288	6.37
*AhARF1*	*AhARF31*	0.0116	0.0166	0.6989	2.04
*AhARF18*	*AhARF46*	0.0349	0.0571	0.6102	7.03
*AhARF17*	*AhARF47*	0.0083	0.0169	0.4911	2.08
*AhARF26*	*AhARF59*	0.0141	0.0304	0.4641	3.74
*AhARF6*	*AhARF37*	0.0295	0.0644	0.4585	7.93
*AhARF28*	*AhARF61*	0.0045	0.0107	0.4234	1.32
*AhARF12*	*AhARF43*	0.0132	0.0320	0.4110	3.94
*AhARF16*	*AhARF48*	0.0165	0.0416	0.3973	5.12
*AhARF15*	*AhARF49*	0.0097	0.0323	0.3015	3.98
*AhARF25*	*AhARF58*	0.0138	0.0461	0.2984	5.68
*AhARF40*	*AhARF61*	0.1741	0.6958	0.2502	85.69
*AhARF28*	*AhARF40*	0.1844	0.7419	0.2486	91.36
*AhARF37*	*AhARF56*	0.1638	0.6923	0.2367	85.25
*AhARF19*	*AhARF50*	0.0049	0.0206	0.2362	2.54
*AhARF36*	*AhARF57*	0.1618	0.7533	0.2148	92.77
*AhARF14*	*AhARF45*	0.0077	0.0442	0.1741	5.44
*AhARF5*	*AhARF36*	0.0025	0.0150	0.1645	1.85
*AhARF39*	*AhARF58*	0.1335	0.8681	0.1538	106.91
*AhARF11*	*AhARF42*	0.0023	0.0188	0.1234	2.32
*AhARF2*	*AhARF33*	0.0025	0.0311	0.0805	3.83
*AhARF22*	*AhARF56*	0.0022	0.0273	0.0793	3.36
*AhARF8*	*AhARF38*	0.0034	0.0525	0.0647	6.47
*AhARF9*	*AhARF40*	0.0013	0.0204	0.0617	2.51
*AhARF20*	*AhARF51*	0.0005	0.0329	0.0154	4.06

## Data Availability

Data are contained within the article.

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
