# Peer review of "Genome-Wide Identification of Auxin Response Factors in Peanut (Arachis hypogaea L.) and Functional Analysis in Root Morphology"

_ijms, 2022, doi:10.3390/ijms23105309_

Round 1
Reviewer 1 Report
The new much improved version is well suited to get accepted to IJMS.
Author Response
Thank you very much for your work and suggestion on our manuscript.
Reviewer 2 Report
The author of the manuscript entitled “Genome-wide identification of auxin response factors in pea- 2 nut and functional analysis in root morphology” is well illustrated and an important to undestant the regulatory mechanisms of ARFs in peanuts and crops. There are some serious errors in the manuscript which require correction prior to its final acceptance.
Comments
Title: write the scientific name of pea plant
(Arachis hypogaea L.)
Introduction
Line 59, 84: check the scientific name. Write in italic form
Peanut (Arachis hypogaea L.)
Line 84, 86: A. hypogaea, A. duranensis and A. ipaensis.
Line 114: remove “although encodes”
Figure 2 , Figure 3, Figure 4. are not clear due to font size. Improve it
Results: The author has used several citations in the result section of the manuscript.
If you are keeping the discussion section separately, no need to use references in the result section.
Discussion: Avoid using figures and tables in the discussion section as you have already illustrated it in the results section.
Author Response
Comments
Title: write the scientific name of pea plant
(Arachis hypogaea L.)
Introduction
Line 59, 84: check the scientific name. Write in italic form
Peanut (Arachis hypogaea L.)
Line 84, 86: A. hypogaea, A. duranensis and A. ipaensis.
Line 114: remove “although encodes”
Response:
Thank you very much for your comments. We have added the scientific name of peanut (Arachis hypogaea L.) in the title (page 1 line 3), write the scientific name in italic form (page 2 line 60, line 85, line 87) and removed “although encodes” (page 3 line111).
Figure 2 , Figure 3, Figure 4. are not clear due to font size. Improve it
Response:
Thank you very much for your suggestion. We have replaced Figure 2, Figure 3, Figure 4 with higher resolution images to ensure the clarity after zooming in.
Results: The author has used several citations in the result section of the manuscript.
If you are keeping the discussion section separately, no need to use references in the result section.
Response:
Thank you very much for your suggestion. In this revised version, we only kept citations helping us describe our results.
Discussion: Avoid using figures and tables in the discussion section as you have already illustrated it in the results section.
Response:
Thank you very much for your suggestion. We have removed the figures and tables in the discussion section.

Round 2
Reviewer 2 Report
Comments
- Results: adjust the citations mentioned in the result section into the discussion section of the text. According to the journal format, citation of previous studies can only be mentioned in the discussion section,
See the author’s instructions as given below.
- Results: Provide a concise and precise description of the experimental results, their interpretation as well as the experimental conclusions that can be drawn.
- Discussion: Authors should discuss the results and how they can be interpreted in perspective of previous studies and of the working hypotheses. The findings and their implications should be discussed in the broadest context possible and limitations of the work highlighted. Future research directions may also be mentioned. This section may be combined with Results.
Author Response
Thank you very much for your comments. We have adjusted the citations mentioned in the result section into the "Discussion" or "Materials and Methods" section of the text.
This manuscript is a resubmission of an earlier submission. The following is a list of the peer review reports and author responses from that submission.
Round 1
Reviewer 1 Report
Luo et al, did a comprehensive in silico study to identify all the Auxin response factors (ARFs) in a common tetraploid peanut cultivar and its two diploid ancestors. Later they identified the potential target genes by scanning the whole genome for Auxin response element or AuxRE. The AuxRE search identified multiple genes involved in root development which can potentially be regulated by ARFs (positively or negatively). Finally, the authors screened a few known peanut varieties phenotypically and made a ARF expression profile on those varieties (to find a co-relation between ARF expression and root architecture). Overall, the study is comprehensive but I am afraid it lacks novelty. There are multiple studies that have shown similar screens and even more studies have shown the connection between ARFs and root development (in multiple model systems).
- The ARF and root architecture studies doesn't really say anything about the role of ARFs in root development (choose specific ARF for a few cultivars and compare expression profiles and root architecture).
- Missing knockout/silencing or overexpression data for any ARF in peanut root (lack of gene expression studies).
- No mechanistic role of any ARF in root development. How is that ARF different or similar to from its counterpart in other model systems (lack of comparative analysis).
- I think at least some localization, gene function and phenotypic study (loss or gain of function) of a couple of ARF proteins would have made this study a lot more interesting.
Reviewer 2 Report
The authors of the manuscript “Genome-wide identification of auxin response factors in peanut and functional analysis in root morphology” have attempted and performed an illustrious work, important to the scientific world in the field of auxins response factors. There are some small issues where the author can concentrate while revising the manuscript prior to its final acceptance.
Comments
Figure1. Figure 1 is not clear. Keep the scientific name of plants in italic form.
“Figure 1. Chromosome distribution of ARFs in cultivar (A) and wild (B) peanuts. Chromosomes belong to the A subgenome of A. hypogaea (Arahy.02-10) and their homologues from A. duranensis were colored in yellow, while the chromosomes that belong to the B sub-genome of A. hypogaea (Arahy.12-20) and their homologues from A. ipaensis were colored in green. Chromosomes do not contain ARFs and ARFs located on the scaffold were not shown”
Figure 5. Check the spelling of “genome” in the following statement.
“Gnome-wide prediction of ARF-binding elements. (A) Definition of DR5 and IR8 elements. N indicates A, C, G, or T. (B) Number of DR5, IR8, and AuxRE in peanut genomes. (C) GO analysis of genes carrying DR5, IR8, or AuxRE in their promoters, and only the GO terms those containing at least 40 genes were shown here”.
- Materials and Methods.
Give detailed growth conditions, area of collection of A. hypogea L. cultivars.